# Complexity of resting cortical activity predicts neurophysiological responses to theta-burst stimulation but fails to generalize: A rigorous machine-learning approach

Matthew Herbert Ning[1], Haoqi Sun[2], Brice Passera[1], Duygu Bagci Das[1], Brandon Westover[2], Alvaro Pascual-Leone[3], Emiliano Santarnecchi[4], Mouhsin M. Shafi[1], Recep A. Ozdemir[1]*

1 Berenson-Allen Center for Noninvasive Brain Stimulation, Beth Israel Deaconess Medical Center, Boston, Massachusetts, United States of America, 2 Neurology Department, Beth Israel Deaconess Medical Center, Boston, Massachusetts, United States of America, 3 Hebrew SeniorLife Center, Boston, Massachusetts, United States of America, 4 Radiology Department, Massachusetts General Hospital, Boston, Massachusetts, United States of America

* rozdemir@bidmc.harvard.edu

## Abstract

### Background

Substantial variability in individual responses to intermittent theta-burst stimulation (iTBS) limits its clinical efficacy, yet neurophysiological mechanisms underlying this variability remain unclear. While most machine-learning studies have focused on modeling behavioral or clinical effects of repetitive transcranial magnetic stimulation (rTMS), the few studies examining neurophysiological outcomes utilized limited feature sets in single-visit settings, which captured only inter-subject variability and most importantly lacked independent validation sets.

### Methods

To address these gaps, we employed supervised machine learning models that integrated baseline resting-state EEG (rsEEG) features and baseline transcranial magnetic stimulation (TMS)-evoked measures, including motor-evoked potentials (MEPs) and TMS-evoked potentials (TEPs), to predict neurophysiological responses to a single iTBS session applied over the primary motor cortex in two independent test-retest studies of healthy adults. We also employed statistical and reliability analysis to understand the statistical relationship between resting state EEG and responses to iTBS.

### Results

Internal cross-validation within the training cohort yielded promising binary classification performance (accuracy: 81%), identifying coarse-grained multiscale distribution

**Data availability statement:** The source code used to produce the results and analyses presented in this manuscript are available from GitHub repository: https://github.com/NoPenguinsLand/TMS-EEG-Machine-Learning-PLOS-CompBio. All relevant data (including resting state EEG features, baseline TMS-EEG, baseline MEPs and outcome measures) are available at https://zenodo.org/records/19238430.

**Funding:** ES was partially supported by the NIH grant No. P01 AG031720 and ADDF (Alzheimer's Drug Discover Foundation) grant No. ADDF-FTD GA201902–2017902. MMS was partly supported by the Football Players Health Study at Harvard University, and the NIH grant Nos. R01MH115949, R01AG060987, R01EB032820, and P01 AG031720. AP-L was partly supported by the NIH grant Nos. R01AG076708, R01AG059089, R03AG072233, and P01 AG031720, the BrightFocus Foundation, and the Barcelona Brain Health Initiative (Institute Guttmann, Barcelona, Spain). The funders had no role in study design, data collection and analysis, decision to publish, or preparation of the manuscript.

**Competing interests:** I have read the journal's policy and the authors of this manuscript have the following competing interests: E. S. serves on the scientific advisory boards for BottNeuro, which has no overlap with present work; and is listed as an inventor on several issued and pending patents on brain stimulation solutions to diagnose or treat neurodegenerative disorders and brain tumors. A. P-L is a co-founder of Linus Health and TI Solutions AG which have no overlap with present work. He serves on the scientific advisory boards for the ACE Foundation and the IT'IS Foundation, Neuroelectrics, TetraNeuron, Skin2Neuron, MedRhythms, and Magstim Inc; and is listed as an inventor on several issued and pending patents on the real-time integration of noninvasive brain stimulation with electroencephalography and magnetic resonance imaging, applications of noninvasive brain stimulation in various neurological disorders, as well as digital biomarkers of cognition and digital assessments for early diagnosis of dementia. B.W. is a co-founder, scientific advisor, consultant to, and has personal equity interest in Beacon Biosignals. None of the other authors

entropy of rsEEG as the most robust predictor of local cortical excitability changes indexed by the 100–131ms window of TEPs. However, predictive performance markedly declined upon external validation (accuracy: 69%), reflecting unstable relationships between predictors and outcomes likely driven by substantial intra- and inter-individual variability of iTBS-induced changes in neurophysiological outcomes.

## Conclusions

These findings emphasize that while EEG complexity measures can capture baseline brain states relevant for neuromodulation to a certain degree, the inherent instability of single-session iTBS effects significantly constrains model generalizability and underscores the necessity of test-retest paradigm to avoid overly optimistic performance estimates. Future studies with multi-session and individualized stimulation protocols are urgently needed to better characterize neurophysiological mechanisms underlying rTMS effects and ultimately enhance its therapeutic potential.

## Author summary

Repetitive transcranial magnetic stimulation (rTMS) is a promising non-invasive neuromodulation technique approved by FDA to treat medication-resistant depression, obsessive-compulsive disorder and smoking addiction, with active research for potential treatment of anxiety, bipolar II disorder and improve post-stroke motor rehabilitation. It's also used experimentally to modify brain excitability, neural plasticity and behavior. However, it currently suffers from low inter- and intra-individual reliability, with some individuals showing improvement from rTMS while others don't. To better understand the underlying mechanism as well as potentially improve its clinical efficacy, we developed a machine learning model that can identify neurophysiological features that will distinguish people who demonstrates cortical target engagement to rTMS apart from those who don't. In order to capture both inter- and intra-individual variability, our participants completed identical rTMS protocols twice, initial session for the first time and retest session for the second time. Our results suggested that the relationship between features and rTMS responses changed over time, limiting our model's ability to generalize. We finally concluded that single session of rTMS isn't effective and suggested that multiple sessions with personalized rTMS parameters are needed to show reliable neurophysiological effects.

## Introduction

Repetitive transcranial magnetic stimulation (rTMS) is a non-invasive neuromodulation technique widely used clinically and experimentally to modify brain excitability, neural plasticity and behavior [1,2]. Among rTMS protocols, intermittent theta burst

report any conflicts of interest. All the other co-authors fully disclose they have no financial interests, activities, relationships and affiliations. The other co-authors also declare they have no potential conflicts in the three years prior to submission of this manuscript.

stimulation (iTBS) has gained increasing attention due to its relatively shorter duration, lower stimulation intensity, and prolonged effects [3]. Furthermore, it was cleared by FDA to treat depression [4] and has the potential to enhance motor recovery after stroke [5] and improve cognitive performance [6]. Despite these promising outcomes, iTBS has limited efficacy owing to substantial inter- and intra-individual response variability [7–9]. Currently, the mechanisms driving this variability remain poorly understood [10].

Based on the results of invasive repeated electrical stimulation studies in animals, theta-burst stimulation (TBS) protocols are originally considered to modulate behavioral responses by altering neural excitability through Hebbian-like synaptic plasticity mechanisms (cortical excitability hypothesis) [11]. This cortical-excitability framework has strongly influenced mechanistic interpretations in humans and, by extension, clinical rationales for TBS, yet accumulating evidence shows significant inter- and intra-individual variability in corticospinal and cortical responses [7–9] and many studies failed to demonstrate consistent neurophysiological effects beyond sham controls [12]. One proposed explanation is that neuromodulatory effects depend on the brain's intrinsic state at the time of stimulation. Optical imaging and electrophysiological recordings in animal models suggest that although evoked responses can be deterministic, variability often arises from the dynamics of ongoing cortical activity [11]. Human transcranial magnetic stimulation combined with EEG (TMS-EEG) studies similarly show that resting-state EEG (rsEEG) features prior to stimulation correlate with variability in rTMS outcomes, emphasizing the role of intrinsic cortical oscillations in shaping responses to transcranial magnetic stimulation (TMS) [12–16]. Importantly, recent evidence suggests that specific features extracted from rsEEG can predict individual differences in corticospinal excitability [17] and may reflect both deterministic and dynamic ongoing neurophysiological characteristics that modulate response to TMS [18].

To date, the majority of studies aiming to characterize inter-individual variability in response to TBS protocol have predominantly focused on predicting behavioral and clinical outcomes using rsEEG derived features. A recent meta-analysis of EEG-based predictive models [19] revealed marked inconsistency across individual studies, reporting highly variable accuracies (approximately 60–90%) and showing that no single EEG-based biomarker has been consistently replicated or validated. While predicting clinical response is essential, it often bypassed the underlying physiological mechanisms. Given that clinical efficacy is assumed to depends on whether the intended neurophysiological mechanism was engaged, focusing solely on clinical outcomes without characterizing the intermediate physiological responses leaves a critical gap in our understanding. If the expected neurophysiological modulation fails to occur due to the individual's baseline state, the subsequent clinical benefit may be compromised. As such, neurophysiological outcomes such as motor-evoked potentials (MEPs) and TMS-evoked EEG potentials (TEPs) provide more proximal indices of physiological "target engagement" and may offer a mechanistic bridge between baseline brain state, stimulation, and downstream clinical or behavioral changes. Predicting these outcomes allows us to test whether

baseline electrophysiological state explains variability in iTBS-induced physiological change, identify individuals likely to show target engagement before lengthy treatment courses, and motivate principled individualization of stimulation parameters to improve efficacy. However, relatively few studies have attempted to predict neurophysiological outcomes, typically using changes in corticospinal excitability as outcome measures [13–15]. Most existing approaches have relied on linear regression analyses linking baseline motor evoked potential (MEP) amplitudes to post-stimulation changes, which may fail to capture the nonlinear and state-dependent nature of cortical plasticity [16–19]. To the best of our knowledge, only two recent studies have applied machine learning models to baseline neurophysiological features such as MEP amplitudes [20] or resting-state EEG-derived spectral power and complexity measures [21]. Although both studies reported promising internal validation accuracy (76–91%), they lacked external validation using independent cohorts, relied on single-session data capturing only inter-individual variability, and limited their outcomes to corticospinal excitability. These methodological constraints raised concerns about overfitting and generalizability with limited applicability beyond the motor cortex. Moreover, given known limitations of MEPs and the promise of TMS-evoked EEG potentials (TEPs) as direct cortical readouts, it is important to test whether rsEEG features predict iTBS-induced changes in both corticospinal and cortical measures.

Here we evaluate baseline predictors of iTBS-induced neurophysiological change using a test-retest design across two independent cohorts. Predictors included both conventional frequency-band spectral power metrics and complexity measures of rsEEG, which have the potential to capture the brain's state-dependency on stimulation [22,23]. Driven by both spectral and nonlinear dynamic components, complexity measures can detect subtle changes in the EEG signal that conventional spectral properties might miss [24,25], and have frequently shown equivalent or superior performance to spectral power metrics in a broad range of EEG applications [24,26–33]. Unlike previous approaches, we examined both MEP as measure of corticospinal excitability and TEP as measure of cortical excitability. For our predictive models, we examined changes in corticospinal and cortical excitability after the iTBS protocol as the response variables. For the corticospinal excitability, the iTBS response is classified as a responder or a non-responder. For the cortical excitability, the iTBS response is classified as facilitation or suppression. We performed statistical tests and reliability analyses to assess the relationship between the predictors and the corticospinal and cortical responses to iTBS protocol. Critically, we trained our predictive models on one cohort and tested the selected model's performance on the second cohort to test generalization performance and leverage a cross-session paradigm to characterize both inter- and intra-individual variability.

## Results

### Analysis sample

After filtering participants with clean rsEEGs, MEPs and TEPs, the analysis sample consists of 21 participants from Cohort 1, 15 of whom have both initial test and retest sessions, 2 of whom only have the initial test sessions and 4 of whom only have the retest sessions, for a total of 36 sessions, and 19 participants from Cohort 2, 18 of whom have both test and retest sessions and 1 of whom only have the retest session, for a total of 37 sessions. Thus, there is a total of 73 sessions in the analysis sample, with 35 initial test sessions and 38 retest sessions.

### Cross-session experiments

Models trained on rsEEG complexity features initially showed promising internal cross-validation performance. The best-performing MEP model (LDA using composite multiscale distribution entropy) achieved a mean ROC-AUC of $75.0 \pm 6.9$, while the top LMFP model (logistic regression with Lasso regularization) reached an ROC-AUC of $83.8 \pm 7.4$ (S4 Fig). However, when tested on the external validation set (independent sessions), performance for both outcomes dropped substantially to near chance levels (ROC-AUC 53.6 for MEP and 48.6 for LMFP, S4 Fig). Please refer to S1 Text for detailed results of the Cross-Session Experiments for both MEP and LMFP.

## Test of inequality in distributions

As a post-hoc analysis to assess the poor performance of the Cross-Session Experiments, we first assessed for the presence of univariate covariate shift in rsEEG by running univariate two-sample Kolmogorov-Smirnov tests between initial test sessions and retest sessions for each EEG channel and feature. Only 2.2% (5/232) of the spectral features have significant differences (without correction) in empirical distribution functions between initial test and retest sessions (see Fig 1 for a representative channel and band). Similarly, only 3.8% (33/870) of the temporal complexity features of rsEEG have significant differences (without correction) in empirical distribution functions between initial test sessions and retest sessions. Both results suggest that their probability distributions remain stable across visits for the majority of the features.

To assess for the presence of label shift in the modulations of iTBS protocol, due to small sample size, two-tailed Fisher exact tests ($\alpha = 0.05$) were run between initial test sessions and retest sessions. With respect to the LMFP ratios, 4 out of 43 (9.3%) different windows have statistically different class distributions (no correction), and all of them occurred after 200ms post TMS-pulse (see the bottom row of Fig 1 for a representative LMFP window). In both T5 and T25 blocks, the class proportions of MEPs didn't differ significantly (T5 Block: P-value = 1.000; T25 Block: P-value = 0.079). This suggests that the probability distributions remain stable across visits for the majority of categorization methods.

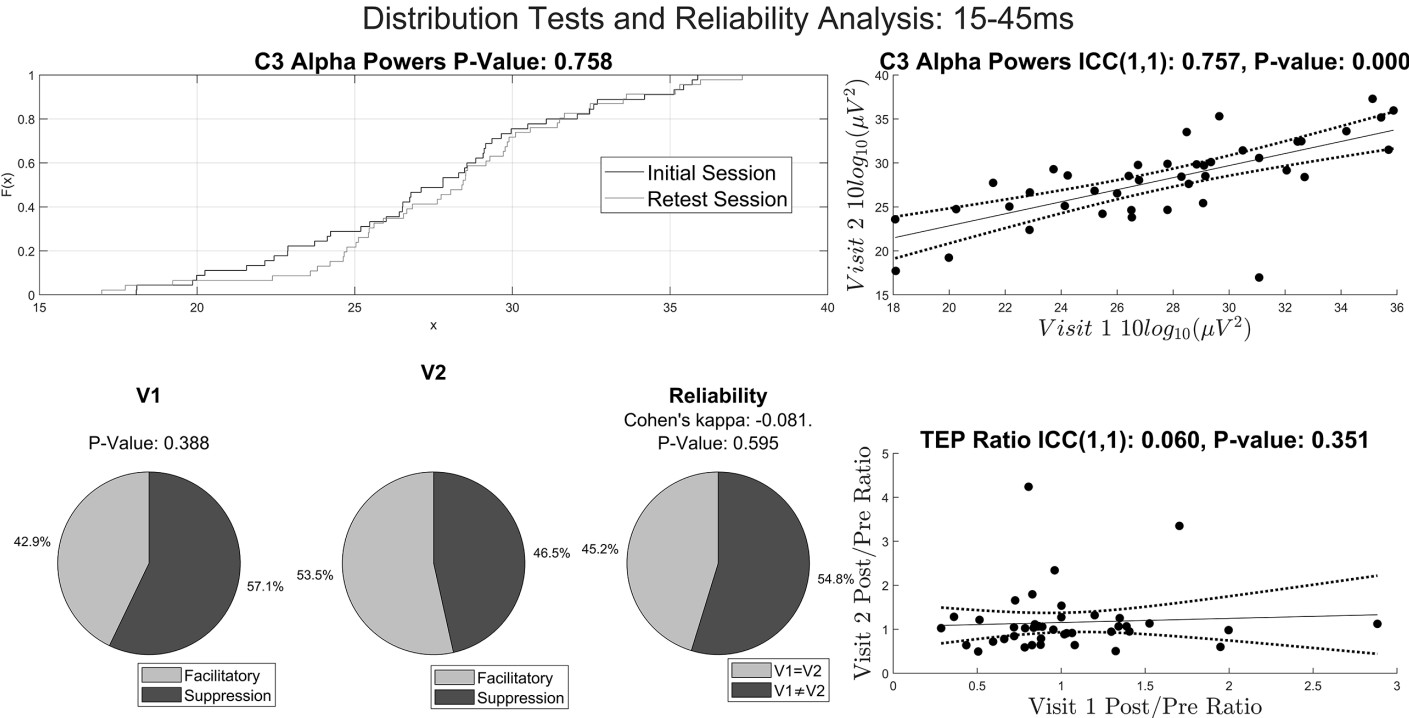

**Fig 1. Distribution Tests and Reliability Analysis.** A) Empirical distribution functions of the alpha band powers from rsEEG from C3 channel from the initial test sessions (dark gray) and retest sessions (light gray) are shown. Alpha band power from C3 channel is representative of the entire sample. The p value represents the Kolmogorov-Smirnoff test statistic. B) Each pie chart in the left and middle columns displays the class distribution whereas each pie chart in the right column displays the percentage of individuals having consistent outcome measures of iTBS-induced neuromodulation across 2 sessions. The top row represents class distribution determined by the ratio of LMFP of the left motor region for the initial test session (left) and retest session (middle). The 15-45 ms window is picked at random. Similarly, the middle and bottom rows represent the MEP T5 and T25 experiments, respectively. The p values in the left column represent the Fisher's exact test statistics whereas the p-values in the right column are computed using the z-tests for Cohen's kappa values. C) Scatter plots with intraclass correlation coefficient (ICC) type (1,1) for the alpha band power from C3 channel (top), LMFP ratio using 15-45 ms window (second from top), p-values of t-tests for MEP T5 block (third from top) and p-values of t-tests for MEP T25 block (bottom). In all 4 cases, the p-values are computed using the F-test. Data interpretation in the main text.

## Reliability analysis

As another post-hoc analysis to assess the poor performance of Cross-Session Experiments, we performed reliability analysis here. The baseline rsEEG band powers generally have high intraclass correlation coefficients (ICC) across visits (mean±SD ICC=0.83±0.11, no correction, see Fig 1 top-right panel for a representative channel and band). Similarly, the temporal complexity of baseline rsEEG have slightly lower but still high reliability across visits (mean±SD ICC=0.62±0.24, no correction). Out of 4 different pre-TBS features, only the AUC of the LMFP of the left motor region has ICC above 0.3 (ICC=0.36, P-value=0.008; regression quality score: ICC=0.09, P-value=0.288; MEP mean amplitude: ICC=0.06, P-value=0.338; MEP standard deviation: ICC=0.03, P-value=0.417).

The reliability of modulatory effects of iTBS protocol across visits remains low, with the ICC of the MEP T5 -0.01 [-0.30, 0.29], MEP T25 -0.27 [-0.53, 0.03] and the mean ICC of the LMFP ratios across 43 different windows 0.05±0.11 (Fig 1 bottom-right for representative LMFP window). Similarly, Cohen's kappa results suggested low reliability with the mean Cohen's kappa of the LMFP ratios across 43 different windows 0.015±0.127 (Fig 1, bottom-left). Similarly, Cohen's kappa for MEP T5 is -0.004 and for MEP T25 is -0.238. The high stability of the probability distributions of spectral powers, different measures of temporal complexity of rsEEG and measures of modulation of corticospinal and cortical excitability and the low reliability of the modulatory effect of iTBS protocol collectively suggested concept drift for the poor external validation performance.

When computing the percentage of individuals with consistent corticospinal and cortical responses across visits for a fixed measure of iTBS-induced modulation, the average percentage across 43 different windows of LMFP ratios is 50.6±6.7% (see Fig 1 top right for a representative window for LMFP Ratio Experiment). 52.3% and 38.1% of individuals have consistent outcomes across visits for MEP T5 and T25 Experiments, respectively.

## Cross-cohort experiment

For the Cross-Cohort Experiments, both initial test and retest sessions from Cohort 1 are included in the training set for the model selection step and initial test and retest sessions from Cohort 2 are included in the external validation set for the model validation step. Additionally, we added a feature to encode the visit type as Initial Session or Retest Session.

For MEPs, linear discriminant analysis with Oracle Shrinkage Approximation trained on the complexity indices of composite multiscale permutation entropy (using normalization) from all EEG channels, using T5 t-test as the categorization method, has the highest cross-validated ROC-AUC (mean±95% confidence interval: 71.4±4.6, accuracy: 71.5±4.5, sensitivity: 70.8±7.6, specificity: 72.0±9.9, precision-recall AUC: 77.4±4.2) (Fig 2 and Table 1) in the model selection step. When tested on validation cohort, the performance slightly fell in all metrics (ROC-AUC: 64.6 [44.9, 79.6], accuracy: 64.7 [52.9, 76.5], sensitivity: 68.4 [47.4, 84.2], specificity: 60.0 [40.0, 80.0], precision-recall AUC: 71.7 [58.5, 85.4]) (Fig 2 and Table 1). The mean performances are above chance level for all metrics. However, the confidence intervals for all metrics contain chance-levels. EEG channels C1, P5 and P1 have the largest coefficient magnitudes in the selected model (S5 Table). The model is relatively large, with 63 features. Coupled it with small drop in performance, the model may still be overfitting to the internal validation set.

For LMFP Ratios Experiment, the model with the highest cross-validated ROC-AUC is the decision tree trained on the complexity indices of coarse-graining multiscale distribution entropy (using Log-Distance transformation) from the left motor region, with the participants classified using the LMFP ratios computed using the 100–131 ms window (mean±95% confidence interval: ROC-AUC: 80.2±4.6, accuracy: 80.6±4.3, sensitivity: 82.0±6.7, specificity: 78.3±9.2, precision-recall AUC: 78.9±4.8) (Fig 2 and Table 1). When tested on the external validation cohort, all performance metrics are above chance-levels (ROC-AUC: 61.1 [49.0, 78.8], accuracy: 69.4 [62.4, 79.2], sensitivity: 77.8 [70.4, 87.1], specificity: 44.4 [22.2, 77.8], precision-recall AUC: 79.5 [74.6, 88.2]) (Fig 2 and Table 1). The confidence intervals of accuracy, sensitivity and precision-recall AUC are above chance-levels whereas the confidence intervals of specificity and ROC-AUC

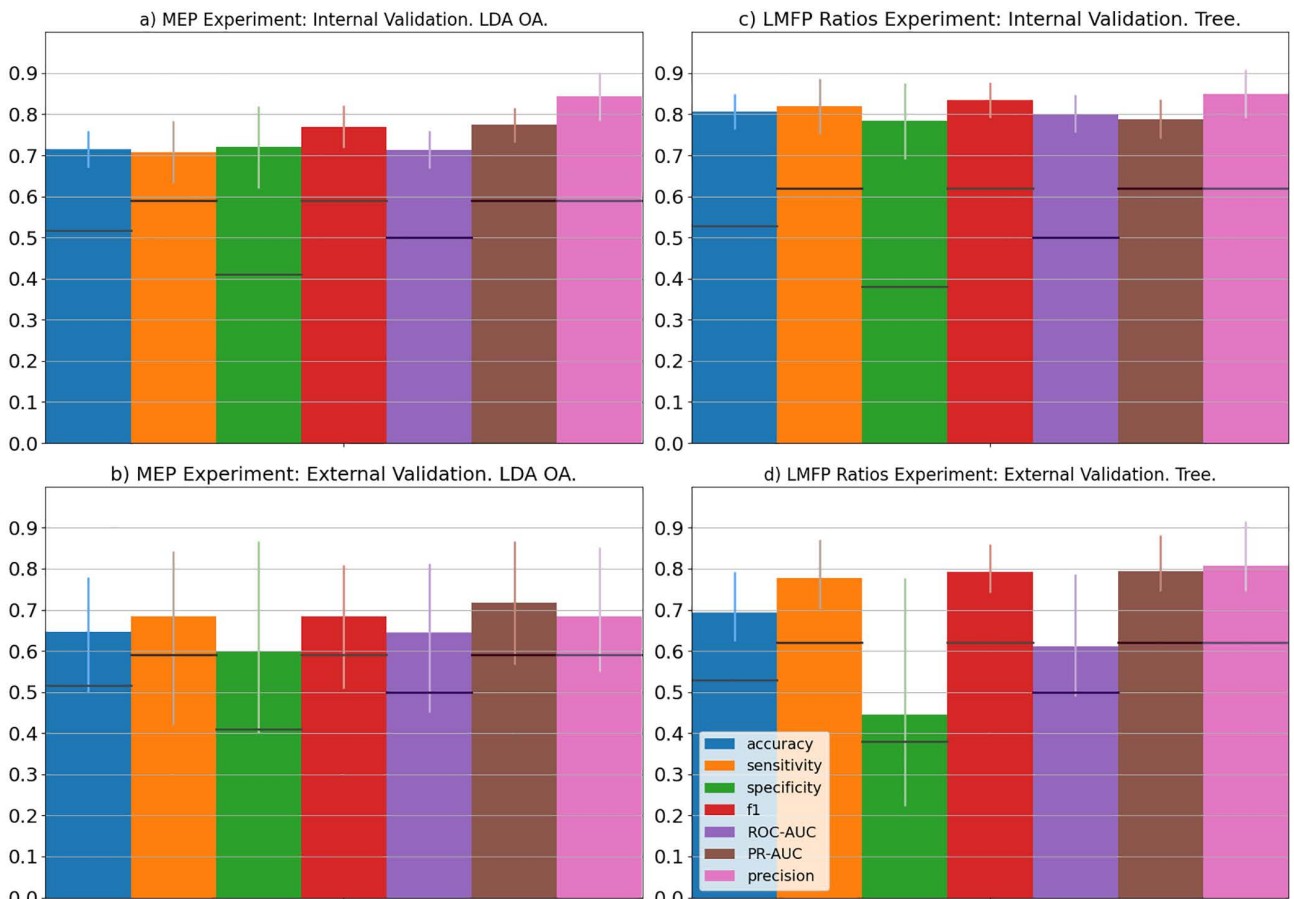

**Fig 2. Model Selection and Model Validation Performance of MEP and LMFP Ratios Cross-Cohort Experiments.** The top row represents the final models selected during the model selection process (as assessed by cross-validation using ROC-AUC metric), bottom row represents the performance of the final models on validation set during model validation. Left column represents the MEP Experiment whereas the right column represents the LMFP Ratio Experiment. The final model for the MEP Experiment is linear discriminant analysis with Oracle approximate shrinkage (LDA OA) trained on the complexity indices of composite multiscale permutation entropy from all EEG channels whereas the final model for the LMFP Ratios Experiment is decision tree trained on complexity indices of coarse-graining multiscale distribution entropy from EEG channels in the left motor region. The vertical error bars represent 95% confidence intervals and the dark horizontal bars within each vertical bars represent theoretical chance levels. 7 different metrics are assessed: accuracy (blue), sensitivity (orange), specificity (green), F1-score (red), ROC-AUC (purple), PR-AUC (brown) and precision (pink).

**Table 1. Performance of model selection and model validation of Cross-Cohort Experiments. Square brackets represent 95% confidence interval. LDA OA stands for linear discriminant analysis with Oracle Approximating Shrinkage.**

| Model Name | Accuracy | Sensitivity | Specificity | F1 | ROC AUC | PR AUC | Precision |
|---|---|---|---|---|---|---|---|
| MEP Cross-Cohort Experiment (LDA OA) | | | | | | | |
| **Internal Validation** | 0.72 [0.67, 0.76] | 0.71 [0.63, 0.78] | 0.72 [0.62, 0.82] | 0.77 [0.72, 0.82] | 0.71 [0.67, 0.76] | 0.77 [0.73, 0.82] | 0.84 [0.78, 0.90] |
| **External Validation** | 0.65 [0.53, 0.77] | 0.68 [0.48, 0.84] | 0.60 [0.40, 0.80] | 0.68 [0.51, 0.81] | 0.65 [0.45, 0.80] | 0.72 [0.59, 0.86] | 0.68 [0.55, 0.85] |
| LMFP Ratios Cross-Cohort Experiment (Decision Tree) | | | | | | | |
| **Internal Validation** | 0.81 [0.76, 0.85] | 0.82 [0.75, 0.89] | 0.78 [0.69, 0.88] | 0.83 [0.79, 0.88] | 0.80 [0.76, 0.85] | 0.79 [0.74, 0.84] | 0.85 [0.79, 0.91] |
| **External Validation** | 0.69 [0.62, 0.79] | 0.78 [0.70 0.87] | 0.44 [0.22, 0.78] | 0.79 [0.74, 0.86] | 0.61 [0.49, 0.79] | 0.79 [0.75, 0.88] | 0.81 [0.75, 0.92] |

contain chance-levels. The model is skewed to high sensitivity and low specificity, suggesting that the model erroneously predicted many cases of suppression to be facilitation. Importantly, the gaps between internal and external validation performance decreased relative to the gap between the training and test sets of the Cross-Session Experiment for the LMFP Ratios (gaps for Cross-Cohort LMFP Ratios Experiment: accuracy = 0.12, sensitivity = 0.04, specificity = 0.34, F1 = 0.04, ROC AUC = 0.19, PR AUC = 0, precision = 0.04, gaps for Cross-Session LMFP Ratios Experiment: accuracy = 0.29, sensitivity = 0.08, specificity = 0.42, F1 = 0.13, ROC AUC = 0.35, PR AUC = 0.39, precision = 0.32).

This suggests that eliminating the time-delay between sessions, and thus removing the effect of concept drift, reduces the performance drop, though it does not fully explain the remaining generalization gap. To further characterize potential sources of the remaining performance drop in the cross-cohort experiment, we performed exploratory univariate distribution checks between cohorts. Across baseline predictors, none of the spectral features differed significantly in distribution between cohorts, and only 7.7% of the complexity features differed in distribution between cohorts. With respect to outcome labels, class proportions for MEP suppression versus facilitation did not differ between cohorts for either block. For LMFP ratios, only 3 of 43 time windows showed different class proportions between cohorts. Collectively, these exploratory univariate checks provide limited evidence for large systematic cohort differences in baseline feature distributions or label proportions, suggesting that the remaining generalization gap is unlikely to be explained by simple covariate shift or label shift alone. Since the decision tree is known to be highly unstable and prone to overfitting, a plot of the final decision tree is shown in Fig 3, showing that the maximum depth is 4 and that only 3 out of the 9 features are used for classification, with coarse-graining multiscale distribution entropy computed from C5 channel having the highest Gini importance (0.70), followed by the regression quality scores of the TEPs (0.24) and coarse-graining multiscale distribution entropy computed from FC5 channel (0.05) (S6 Table). Interestingly, visit type isn't used here.

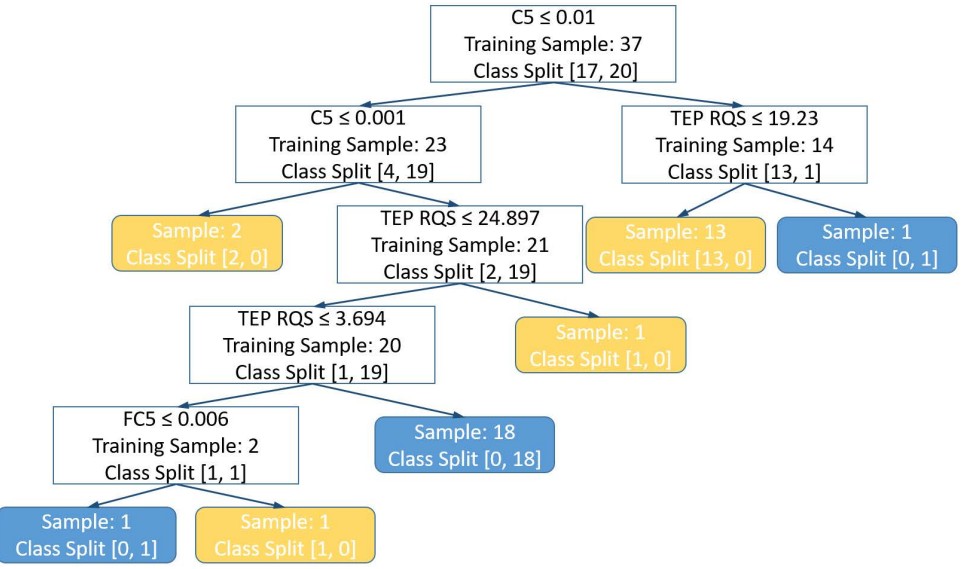

**Fig 3. Plot of the final decision tree from LMFP Ratios Cross-Cohort Experiment.** If the condition in the top row of the white box is true, the decision path takes the left node, or else, takes the right node. "Training Sample" in the white boxes represents the sample size of the training set before the split. "Class split" represents the class ratio of the training set as follows: [decrease in cortical excitability, increase in cortical excitability]. Yellow box represents a decrease in cortical excitability as assessed by the ratio of the AUC of the 100-131 ms window of the left motor region of LMFP. Similarly, blue box represents an increase in cortical excitability. RQS stands for regression quality score.

To examine for possible overfitting, the performances of other models of varying level of complexity including decision tree with a maximum depth of 2 (as opposed to a depth of 4 used in the Cross-Cohort Experiment) trained and tested on the same selected feature set were inspected. In internal validation, only decision tree with a maximum depth of 2 and Gaussian Naïve Bayes have ROC AUC > 0.70 (S1 Fig). Similar to the selected model, the external validation performances of both models (S2 Fig) are skewed with higher sensitivity and lower specificity. For the rest of the models, external validation performances are also skewed toward higher sensitivity or specificity (S2 Fig). Thus, the performance drop is likely due to data bottleneck rather than overfitting. One possible explanation for the data bottleneck is the study-wide bias.

## Cross-subject LMFP experiment

To examine possible study-wide bias, we next randomly sampled 50% of the subjects (including both initial and retest sessions) from both cohorts for the internal validation set and the remaining subjects make up the external validation set. The model with the highest cross-validated ROC-AUC is the decision tree trained on the complexity indices of time-shifted multiscale distribution entropy (using Log-Distance transformation) from the central region, with the participants classified using the 85–115 ms window (mean ± 95% confidence interval: ROC-AUC: 75.0 ± 5.1) (S3 Fig). When tested on the external validation cohort, the performance dropped to chance-levels (ROC-AUC: 50.6 [41.0, 65.3]) (S4 Fig). One possible interpretation is that the drop in performance we observed in Cross-Cohort Experiment is not due to study-wide biases. However, training the model to both cohorts does not necessarily eliminate study-wide biases but can still introduced shift in data distribution, this suggested that study-wide bias possibly caused data bottleneck, limiting external validation performance in both Cross-Cohort and Cross-Subject Experiments.

## Discussion

Given the widespread clinical use of theta-burst stimulation (TBS) and the long-standing assumption that clinical effects of TBS are due to reliable physiological target engagement, we investigated whether baseline electrophysiology can predict the direction of iTBS-induced neurophysiological change and whether such prediction generalizes across independent samples. We trained supervised machine-learning models on comprehensive baseline features capturing both spontaneous (rsEEG) and stimulus-evoked (MEPs and TEPs) brain dynamics to predict the neurophysiological effects of a single iTBS session applied to the primary motor cortex in two independent cohorts of healthy adults. We found that the performance of the LMFP Ratios Cross-Session Experiment in the model selection step is comparable or superior to those of previous machine learning studies [20,21] that only utilized a single visit. Although baseline features remained stable, iTBS-induced changes in neurophysiological outcomes exhibited low test-retest reliability with considerable intra- and inter-individual variability, pointing to concept drift, an unstable predictor-outcome mapping, as the primary barrier to model generalization within the Cross-Session Experiment. Moreover, to test generalization across independent, we split data by cohort for the Cross-Cohort Experiment. Internal cross-validation in Cross-Cohort Experiment achieved accuracies of 72–81% and identified coarse-grained multiscale distribution entropy of the left motor region of rsEEG as the strongest predictor of iTBS-induced changes in local cortical excitability. However, when the selected model was externally validated on the second cohort under near-identical experimental conditions, predictive performance markedly declined, highlighting critical challenges in achieving generalizable models. Crucially, rigorous internal analyses ruled out cohort-specific biases including both covariate and label shifts. Consistent with this, a cross-subject analysis that randomly sampled subjects from both cohorts for training and testing still yielded poor external validation performance, suggesting that the generalization gap is not explained by "cohort identity" alone. Furthermore, post-hoc comparisons revealed that neither simpler linear models nor complex methods could recover the performance. We therefore interpret the Cross-Cohort performance drop as reflecting residual dataset shift and limited effective signal-to-noise for single-session after-effects, rather than concept drift occurring over time.

The marked discrepancy in predictive performance between training and test sets in Cross-Session Experiments highlights the challenges of developing generalizable prediction models for iTBS-induced neurophysiological effects. To better understand the potential source of performance drop in the Cross-Session Experiments, we assessed covariate shift by evaluating the stability of baseline features (rsEEG, MEPs, and TEPs) across visits and label shift by examining potential changes in outcome class distributions. First, using univariate Kolmogorov-Smirnov tests, we found that the distributions of spectral powers and temporal complexity of baseline rsEEG generally stay stable across initial and retest samples, ruling out univariate covariate shift. This is consistent with previous studies [34,35]. Using Fisher's Exact Test, few measures of the modulation of cortical excitability were found to have significantly different class proportions between initial and retest samples and all of them are after 200 ms after the TMS pulse. This ruled out label shifts for the LMFP periods before 200 ms post-TMS-pulse and is consistent with previous study [36]. Importantly, we found low reliability in the modulatory effect of iTBS protocol, as assessed by the intraclass correlation coefficients and Cohen's kappa values. These findings collectively point toward concept drift, an unstable or inconsistent relationship between predictive features and outcomes, as the primary driver of poor generalization. Although we did not directly quantify changes in feature-outcome relationship over time, the stability of baseline predictors and the group-level consistency of outcome classes strongly imply that individual-level variability in response to iTBS underlies the observed drift. In Cross-Cohort Experiment, we observed reduced performance gap relative to Cross-Session Experiment. To examine residual performance gap for possible overfitting, we compared the performance of other classifiers, including another decision tree with shorter maximum depth and found that simpler models does not fix the skewed external validation performance, suggesting that the performance drop are possibly due to data bottleneck. Next, we examined for possible study-wide bias by sampling half of the subjects (including both initial and retest visits) from both cohorts for the internal validation set and the remaining for the external validation set for the Cross-Subject Experiment. We found large performance drop in external validation set. One possible explanation is that study-wide bias does not explain the performance gap seen in the Cross-Cohort Experiment. However, including subjects from both cohorts does not necessarily eliminate study-wide bias, it's also possible that study-wide bias still explains the performance gap in both Cross-Cohort and Cross-Session Experiments.

Data bottleneck could arise for a number of reasons. For instance, our chosen outcome measures (MEPs and TEPs) of cortical excitability may not fully capture the neurophysiological effects of iTBS, suggesting a potential mismatch between measured outcomes and actual neuromodulatory processes. Therefore, the observed poor generalization may partially reflect limitations of our current neurophysiological measures in accurately and consistently indexing iTBS effects across individuals and sessions. The cortical-excitability hypothesis was formulated largely on the basis of early human motor-cortex studies in which single-session rTMS produced group-level changes in MEP amplitudes with high-frequency or patterned rTMS (e.g., iTBS) increasing corticospinal excitability [3] while low-frequency protocols produced the opposite effect [37]. However, these early seminal reports were typically under-powered, lacked robust sham controls, and were not replicated across repeat sessions. Indeed, more recent studies with larger sample sizes and repeat session sham-controlled designs showed that single session of rTMS protocols, including iTBS, do not consistently modulate canonical measures of cortical excitability beyond robust sham protocols in healthy participants and often show poor test-retest reproducibility across identical visits [12]. These reports raise the possibility that MEP and TEP measures may be *insufficient* physiological read-outs for capturing the main biological actions of rTMS. Recent mechanistic reviews indicates that rTMS induces a far richer array of neural changes than simple shifts in excitability. These effects include the widespread modulation of neurotransmitter systems like dopamine [38] and serotonin [39,40], the triggering of activity-dependent gene expression [41,42] and epigenetic remodeling [43,44], the release of neurotrophic factors like BDNF [45,46], and the engagement of broader neuro-endocrine [47–49] and glial pathways [50].

Several alternative explanations should also be considered. One possible explanation is that, even if the cortical-excitability hypothesis remains partially valid, a single session of iTBS may be insufficient to reliably induce measurable changes in MEP or TEP indices. In clinical practice, rTMS treatments typically involve multiple daily sessions over several

weeks, whereas most experimental studies evaluate neuromodulatory effects using a single pre-post stimulation design and track responses over 60–90 minutes. Animal research has demonstrated that different stimulation doses engage distinct neural mechanisms [51]. While a single high-frequency rTMS session primarily affects transient membrane potentials and ionic currents, repeated daily sessions may drive alterations in neurotransmitter receptor levels [52–55] and sustained neurotrophic signaling, such as brain-derived neurotrophic factor elevation [45,46], leading to stable and durable network reorganization. Consequently, the short-lived, activity-dependent plasticity after single-session rTMS protocols likely exhibits considerable variability both within and between individuals, whereas repeated sessions may generate cumulative and more reliable neurophysiological changes. Interestingly, one study showed that multiple blocks of iTBS in a day does not improve reliability [56]. However, that study doesn't use the same volume of sessions spread over multiple days as the clinical studies do. Another potential explanation might be the sensitivity of our features to detect iTBS induced changes. Although our baseline EEG complexity features demonstrated statistical stability, they might lack sufficient sensitivity to detect subtle and transient physiological changes induced by a single iTBS session. While EEG complexity measures, such as approximate entropy and multiscale entropy, effectively distinguish pathological states and predict clinical responses to multi-session rTMS [57], their utility for predicting immediate neurophysiological responses to single-session iTBS has not been systematically evaluated. Thus, it remains possible that they reflect broader, slower network dynamics relevant for clinical outcomes rather than transient, circuit-specific plasticity. Finally, we may be looking at the wrong features. While we intentionally restricted our feature set to spectral power and complexity to maintain statistical robustness and interpretability, other metrics such as dynamic functional connectivity and phase-amplitude coupling (PAC) could offer complementary insights into the brain's receptivity to stimulation and could be explored in future studies. Ultimately, interpreting single-session MEP or TEP shifts as definitive markers of rTMS-induced neuroplasticity may oversimplify or overlook the more complex and multiscale biological processes elicited by repeated stimulation.

Despite poor generalization, there are important observations from the results. Our results revealed that the temporal complexity of rsEEG was selected over spectral properties of rsEEG in the model selection steps of all six experiments, showing that they have higher internal validation performance. This is consistent with previous studies [24,26–33]. Moreover, in all six experiments, multiscale entropies were selected over single scale entropies. In M/EEG studies, different temporal scales of temporal complexity of the brain signal were shown to be linked to different scales of cortical processing, with the structure of variability at short time scales, or high frequencies, linked to local neural population processing, and the variability at longer time scales, or lower frequencies, linked to large-scale network processing [58–60]. Furthermore, we found that the best performance was for the LMFP ratios based on the 100–131ms time window following TMS pulses (Accuracy: 0.81), suggesting that N100 peak responses, a closely related characteristic, may play a role in the modulatory effects of iTBS. The N100 is one of the most robust and widely studied TEP components, consistently elicited from multiple sites of stimulation including motor, prefrontal, parietal, or cerebellar stimulation sites and is tightly linked to GABAergic mediated cortical inhibition [61]. While iTBS protocol to M1 region is well studied [62], reports on the specific effect on N100 are very limited and inconsistent. For instance, one study reported a non-reproducible increase of N100 amplitude [12], while another one found a significant reduction [63]. These heterogeneous results suggest that a single session of iTBS protocol to M1 cortex does not have a reproducible effect on N100 peak and likely explain the performance drop we observed during external validation.

Several limitations should be considered when interpreting our findings. First, although our dataset (73 sessions across 40 participants) is comparable or larger than recent ML studies predicting neurophysiological outcomes from TMS, it may still be insufficient for robustly training predictive models, partially explaining the wide confidence intervals observed in external validation. Second, our sample included only healthy controls, limiting direct generalizability to clinical populations, whose neurophysiological responses to iTBS may differ significantly. In addition, minor methodological differences between cohorts, such as the number of single-pulse TMS trials, post-TBS sampling windows, intervals between sessions, and slight variations in EEG preprocessing, may have introduced additional noise or cohort-specific variability. In addition, our outcome

labeling strategy may have introduced boundary-related label noise for cortical measures: whereas MEP-based responder labels were derived from within-subject statistical testing across trial-level distributions, LMFP/TEP outcomes necessarily relied on ratio thresholding ($\geq 1$ vs $< 1$), which can classify very similar ratios (e.g., 0.99 vs 1.01) into different classes and may reduce discriminative power when effects are graded. Finally, the lack of ground truth in measuring change in corticospinal and cortical excitability presumably induced by the iTBS protocol required us to test multiple different measures. Furthermore, these measures are often noisy and can cause errors in labels, which can weaken the validity of the machine learning experiment even if the predictive performance is strong and require independent validation of proposed measures of iTBS-induced changes. Addressing these issues with larger, clinically diverse samples, standardized protocols, and multimodal outcome measures will be critical to identify robust biomarkers of neurophysiological responses to rTMS protocol.

In summary, our findings demonstrate that while baseline rsEEG complexity measures can predict iTBS-induced changes in local cortical responses to a certain degree, the neurophysiological outcomes derived from single-session protocols are too unstable to support predictive models that generalize reliably beyond the training dataset. The decline in performance observed during external validation underscores the importance of having independent dataset and using a test-retest paradigm to avoid overly optimistic estimates of model accuracy. Overall, the considerable variability in individual responses remains a central barrier to optimizing the efficacy of rTMS. Addressing this challenge may require developing personalized stimulation strategies tailored to each individual's unique brain anatomy and baseline neurophysiology, and systematically evaluating these strategies through multi-session protocols and richer, multimodal biomarkers that can effectively link robust behavioral improvements to their underlying neural mechanisms.

## Methods

### Ethics approval and consent to participate

In accordance with the Declaration of Helsinki, experimental protocols and voluntary participation procedures were explained to all participants before they gave their written informed consent for the study. All questionnaires and procedures were approved by the Institutional Review Board of the Beth Israel Deaconess Medical Center, Boston, MA.

### Studies

This study utilized two independent sham-controlled test-retest reliability studies, both collected at Beth Israel Deaconess Medical Center. Cohort 1 was collected from 2018-2021 [64] and Cohort 2 was collected from 2016-2019 [12,65].

### Participants

For the Cohort 1 Study, data were collected from 28 participants (18 males; mean $\pm$ SD age $= 39 \pm 16$ years). For the Cohort 2 Study, data were collected from 24 participants (16 males; mean $\pm$ SD age $= 30 \pm 11$ years, range $= 18$–49). In both studies, all participants were right-handed (assessed by modified Edinburgh handedness inventory) and none had contraindications to TMS or magnetic resonance imaging (MRI), self-reported history of psychiatric or neurological diseases or evidence of drug abuse. In both studies, the participants were not taking any psychoactive medication at the time of measurements. Additionally, caffeine intake, sleep, and menstrual cycle for females were controlled in both cohorts. In accordance with the Declaration of Helsinki, experimental protocols and voluntary participation procedures were explained to all participants before they gave their written informed consent for the study. All questionnaires and procedures were approved by the Institutional Review Board of the Beth Israel Deaconess Medical Center, Boston, MA.

### TBS procedures

Details of TMS sessions, technical specifications and parameters used, determination of motor hotspot, resting motor threshold (RMT) and active motor threshold (AMT), methods used for TEP and MEP recording, preprocessing and analyses are presented in S1 Text.

In both studies, single-pulse TMS were administered at 120% of the RMT to the motor hotspot, in accordance with the safety guideline and recommendation from International Federation of Clinical Neurophysiology [66]. In Cohort 1, there were three sets of single-pulse TMS blocks; a set of 150 pulses delivered before the rTMS protocol (pre-TBS), another set of 150 pulses at 5 minutes after the rTMS protocol (T5), and a third set of 60 pulses at 25 minutes after the rTMS protocol (T25). In Cohort 2, there was a total of four sets of single-pulse TMS blocks; a set of 120 pulses before the administration of rTMS protocol (pre-TBS), a set of 120 pulses at 5 minutes after the administration of rTMS protocol (T5), a set of 60 pulses at 20 minutes post rTMS protocol (T20) and another set of 60 pulses at 30 minutes post rTMS protocol (T30). In this report, single-pulse TMS from both studies were consolidated to make up the Pre-TBS Set, T5 single-pulse TMS from both studies were consolidated to make up the T5 Set and T20 single-pulse TMS from Study 2 and T25 single-pulse TMS from Study 1 were consolidated to make up the T25 Set (Fig 4, navy circle and arrows).

In both studies, the baseline resting state EEG (eyes-opened) was recorded for three minutes before the application of rTMS protocol but after the baseline single-pulse TMS (Fig 4, light blue circle and arrow). In both studies, the iTBS protocol was applied to the motor hotspot at 80% of the AMT (Fig 4, yellow circle and arrow), also in accordance with the aforementioned safety guideline [66].

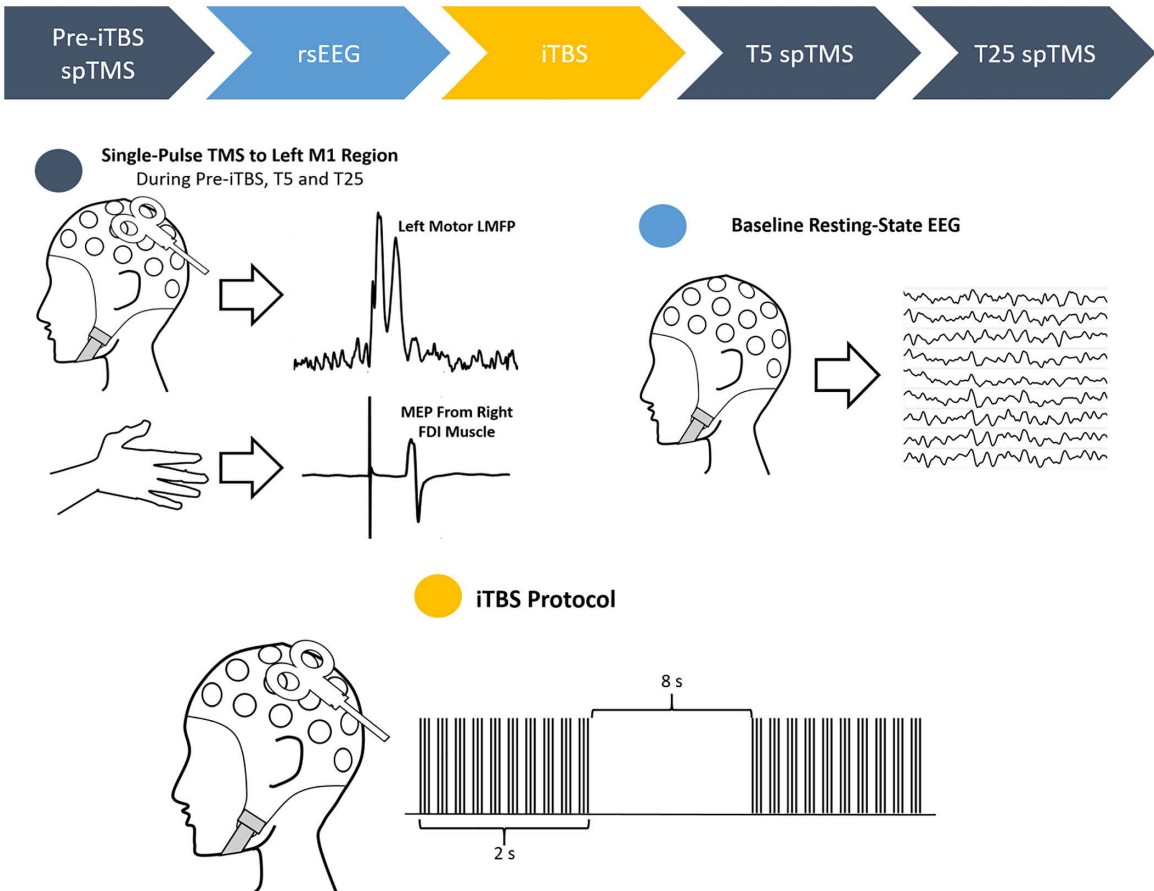

**Fig 4. Diagram of iTBS session, applicable to both initial test and retest sessions.** Top) Chronological order of the session with the color of the blocks corresponding to the color of the circles. Navy circle) Three blocks of single-pulse TMS were delivered to the left M1 region, one before the iTBS protocol, one 5 minutes after and another 25 minutes after the end of the iTBS protocol. Both EEG and EMG were recorded. Light blue circle: 3 minutes of resting-state EEG with eyes open. Only EEG was recorded. Yellow circle: iTBS protocol consisting a total of 600 bursts spread out in an alternating sequence of trains and silence. Only EEG was recorded.

## Outcome measures of iTBS-induced Neuromodulation

As an index of iTBS-induced modulation of corticospinal excitability, we quantified changes in MEP peak-to-peak amplitudes at the individual level. For each session, we conducted a two-tailed, two-sample t-test ($\alpha = 0.05$) comparing the distribution of MEP peak-to-peak amplitudes from the Pre-TBS block with those from the post-stimulation blocks (T5 or T25). A two-tailed test was used to avoid imposing a priori assumptions about the direction of iTBS-induced modulation. Throughout the manuscript, these outcomes are referred to as the **MEP T5 t-test** and **MEP T25 t-test**, respectively. The resulting p-values were used to define individual-level response labels: participants were classified as *responders* if the post-iTBS MEP distribution differed significantly from baseline ($p \le 0.05$), indicating either facilitation or suppression, and as *non-responders* otherwise. For the TEPs, local mean field power (LMFP) of different windows was computed as measures of the strength of local cortical activation following TMS pulses, using the following formula:

$$LMFP(t) = \sqrt{\frac{\left[\sum_i^k \left(V_i(t) - V_{mean}(t)\right)^2\right]}{K}}$$

(1)

where t is time within a given window, $V_i$ is the voltage in channel i, K is the number of channels and $V_{mean}$ is the mean voltage across EEG channels [67]. Windows were either determined a priori based on previous literature [68,69] or systematically tested by looping 30 ms-long window over the 25–345 ms period post TMS pulse. Thus, a total of 43 different windows (S1 Table) were tested in this study. Each window represents a categorization method. The area-under-the-curve (AUC) of the left motor LMFP of each window was computed using the composite Simpson's rule from the following EEG channels in the left motor cortex region: C1, C3, C5, FC1, FC3 & FC5 and were averaged across all trials in each block. Next the ratio is taken between the mean AUCs of Post-TBS (T5 or T25 Set) to Pre-TBS Set:

$$LMFP\ ratio(w) = \frac{AUC\ of\ LMFP(w)_{Post-TBS}}{AUC\ of\ LMFP(w)_{Pre-TBS}}$$

(2)

where w is the window. The modulation in cortical excitability is classified as facilitation if the left motor LMFP ratio for a given window is greater than or equal to 1, and suppression otherwise. These measures of modulation in cortical excitability will be referred throughout here as left motor LMFP ratios, with specific windows specified as needed.

## Statistical tests for covariate shift and label shift

Supervised algorithms typically depend on the distributions of both the underlying features ($\boldsymbol{x}$) and labels ($\boldsymbol{y}$) (responder vs non-responder for MEP t-test or facilitation vs suppression for LMFP ratio) to be independent and identically distributed across samples. However, strict adherence to assumptions rarely arises in real-world scenarios. For post-hoc analysis, to diagnose poor ML performance, we performed statistical tests for covariate shift and label shift on our datasets. By definition, covariate shift occurs when $p(\boldsymbol{x})$ but not $p(y|\boldsymbol{x})$ changed and label shift occurs when $p(y)$ but not $p(\boldsymbol{x}|y)$ changed. To test for univariate covariate shift, two-sample Kolmogorov-Smirnov test ($\alpha = 0.05$) was run for each feature between one sample consisting of the initial test sessions and another sample consisting of the retest sessions. To test for label shift, due to small sample size, two-tailed Fisher's exact test ($\alpha = 0.05$) was run instead of chi-squared test between two samples for each categorization method for both corticospinal and cortical excitability.

## Detecting concept drift

By definition, concept drift occurs when the posterior distribution $p(y|\boldsymbol{x})$ but not $p(\boldsymbol{x})$ changed. There's no formal statistical test for concept drift. Instead, concept drift will be inferred through a combination of aforementioned statistical tests and reliability analysis.

## Reliability analysis

For post-hoc analysis, to assess test-retest reliability, intraclass correlation coefficients (ICCs) based on one-way random effects model (ICC(1,1)) were computed between visits for each band powers, measure of complexity of baseline rsEEG, left motor local mean field power ratios and MEP T5 and T25 p-values. The intraclass correlation coefficient is a measure of correlation between repeated measures within the same individuals or groups. For the one-way random effect model, the ICC(1,1) is computed as followed:

$$ICC(1, 1) = \frac{MS_R - MS_W}{MS_R + (k - 1)MS_W}$$

(3)

where $MS_R$ is the mean square between groups (individuals), $MS_W$ is the mean square for residual sources of variance and k is the number of measurement for a given object of measurement [70]. The p-values of the ICCs are computed using the F-tests (F statistic = $MS_R/MS_W$). Additionally, Cohen's kappas were computed between visits using the classifications based on left motor local mean field power ratios and MEP T5 and T25 t-tests. Cohen's kappa is computed as follow:

$$\kappa = \frac{p_o - p_e}{1 - p_e}$$

(4)

where $p_o$ is the relative observed agreement among multiple measures of a single quantity of interested and $p_e$ is the hypothetical probability of chance agreement based on the following formula:

$$p_e = \frac{1}{N^2 \sum_k n_{k1} n_{k2}}$$

(5)

where k is the number of categories, N is the number of observations and $n_{ki}$ is the number of times ith measure predicted category k. The p-values of Cohen's kappa values are computed using the z-tests. Cohen's kappa value can range from -1–1, endpoints inclusive. Finally, we computed the percentage of individuals with same outcome measures across two sessions (e.g., classified as facilitation using left motor LMFP ratio using window 15–45ms in both initial and retest sessions).

## Machine learning experiments

A machine learning model is a classifier trained on a given pair of feature set (see next four sections) and a measure of modulation in corticospinal or cortical excitability. In Cross-Session Experiment, initial iTBS sessions from both cohorts make up the training set for the model selection process whereas retest iTBS sessions from both cohorts make up the test set. Post-hoc analysis, including statistical tests for covariate and label shift and reliability analysis will be performed on the results of the Cross-Session Experiment. In Cross-Cohort Experiment, sessions from Cohort 1 make up the internal validation set for the model selection process whereas sessions from Cohort 2 make up the external validation set. Additionally, the visit type (initial test session vs retest session) was encoded and included as an additional feature to capture intersession variability in classifications. Finally, to examine possible study-wide bias, in Cross-Subject Experiment, 50% of the subjects (including both initial and retest sessions) are sampled from both cohorts for the internal validation set and the rest for the external validation set. Measures of rsEEG spectral powers, temporal complexity, baseline MEPs and left motor LMFPs were included as features for MEP and left motor LMFP ratio prediction models. Internal validation has the same purpose as training set but emphasizes that all data trained is from one cohort. Similarly, external validation is similar to test set but emphasizes that data tested are drawn from an independent cohort.

## Baseline rsEEG band powers features

Power spectral densities were estimated from baseline rsEEG using the multitaper method (using Discrete Prolate Spheroidal (Slepian) Sequences as tapers) for each EEG channel and within every 10 s epochs. Next, the band powers for the four frequency bands are defined as the area under the curve of the power spectral densities using the following frequency ranges: delta (1–4 Hz), theta (4–8 Hz), alpha (8–12 Hz), beta (12–20 Hz). Finally, the band powers are averaged across epochs for each EEG channel and participant.

## rsEEG entropy features

Different measures of complexity (S2 Table) at fixed temporal scale were extracted from each EEG channel recorded during baseline rsEEG session: approximate entropy [71], sample entropy [72], permutation entropy with embedding dimension 3 [73], distribution entropy [74], incremental entropy [75] and Lempel-Ziv complexity [76]. For multiscale entropy, complexity indices are computed for permutation entropy of embedding dimension 2 and 3, sample entropy and distribution entropy using coarse-graining [77], time-shifted [78] and composite multiscale procedures [79]. Complexity indices are computed as the AUC of the multiscale entropy curve over 20 temporal scales. All complexity measures are computed from a 1-minute recording (first 6 10-s epochs).

## Baseline TMS features

Four different features based on the MEPs and TEPs from pre-iTBS (baseline) block were computed: the mean peak-to-peak MEP amplitude and its standard deviation, the mean AUC of the LMFP of the left motor region using 15–80 ms window after the pulses and the mean regression quality score, a regression-based composite measure of assessing the consistency of individual trials in TEPs [80]. For the MEP Experiments, to capture both local and distal connection, these four features made up the pre-TBS feature set. For the LMFP Ratios Experiments, to capture only the local response, only the mean AUC of the LMFP from the baseline TEPs and regression quality score of TEP made up the pre-TBS feature set.

## ROI

Three different regions of interest are left motor, central and whole-scalp (defined in S3 Table). As band powers and entropies are computed for each EEG channel, a feature group is defined by the ROI, which determine the EEG channels used (Fig 5).

## Classifiers

Nine different types of classifiers were tested in the model selection step: logistic regression with L2 regularization with inner cross-validation for hyperparameter tuning; linear discriminant analysis with inner CV for hyperparameter tuning; linear discriminant analysis using Ledoit-Wolf estimator; linear discriminant analysis using Oracle Shrinkage Approximation estimator; nearest shrunken centroid using Manhattan distance metric; nearest shrunken centroid using Euclidean distance metric; Gaussian Naïve Bayes using empirical priors; Gaussian Naïve Bayes using a priori-defined class priors (80–20 ratio) and decision tree.

Let TP, TN, FP and FN denote the number of true positives, true negatives, false positives and false negatives, respectively. The models were assessed with 7 metrics defined as follows:

$$accuracy = \frac{TP + TN}{TP + TN + FP + FN} \tag{6}$$

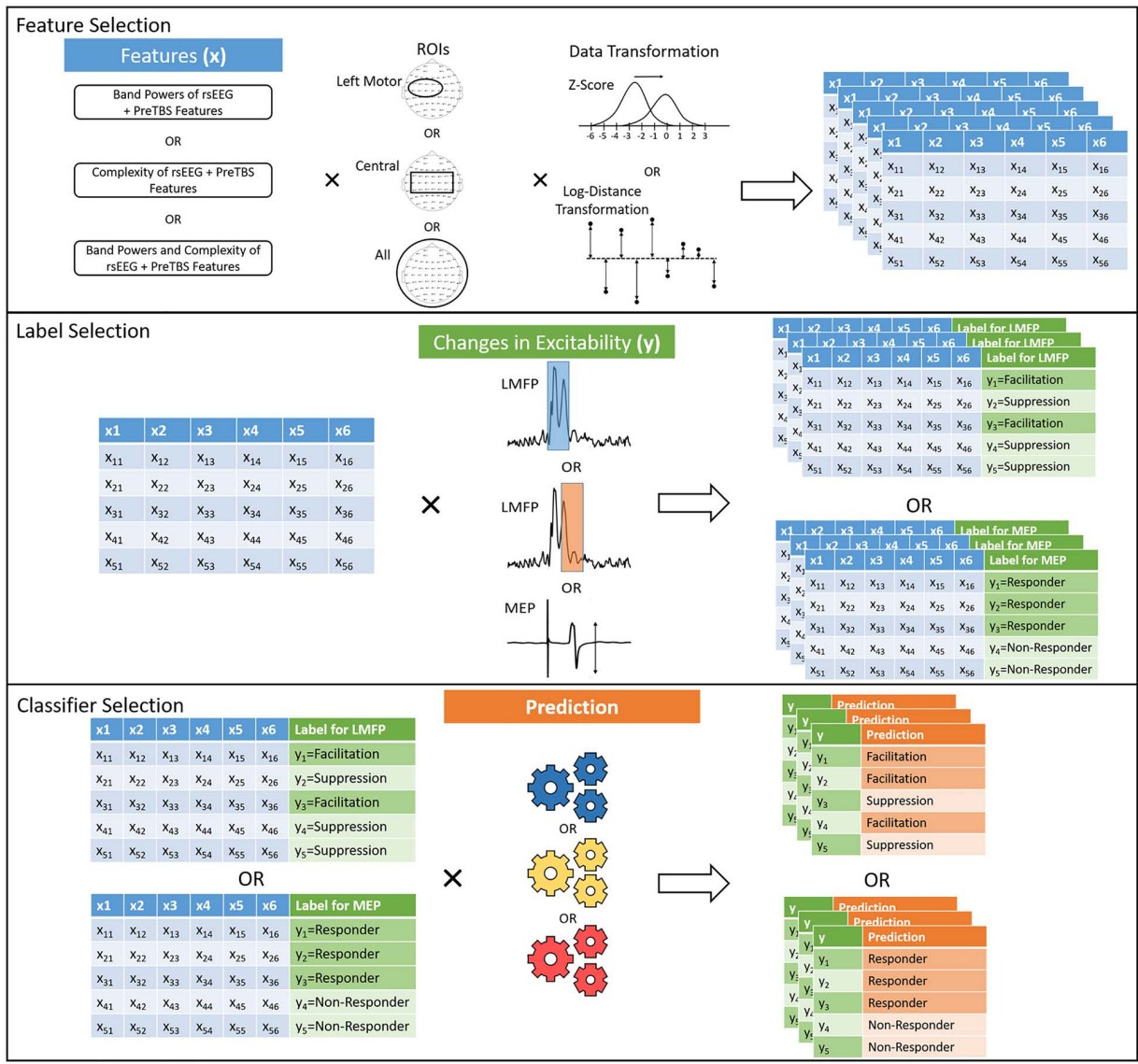

**Fig 5. Schematic Diagram of Feature Grouping and Model Generation.** In the top panel, feature groups were iteratively generated by generating all combinations of different features (S2 Table), ROIs (S3 Table and data transformations (normalization in top or distance to the median in bottom). An example of a feature group would be normalization of the alpha band powers from the central region of interested and PreTBS Features. Each table represents one feature group. In the middle panel, feature groups with labeling of modulation in corticospinal or cortical excitability are further iteratively generated by generating all combinations of feature groups and categorization methods, represented as a green column appended to the feature group table (different windows of LMFP (S1 Table) or t-tests of peak-to-peak MEP amplitudes between post and pre-TBS protocol). In the bottom panel, each feature group with labeling is trained by different classifiers, represented as gears, to form models. An example of a model would be a logistic regression with L2 regularization trained on the same example feature group above to the iTBS responses as facilitation or suppression based on the ratio of the LMFP between post- and preTBS sessions for the 55-85 ms window. See the main text for the total number of models tested in MEP and LMFP Ratios Experiments.

$$sensitivity \ (recall) = \frac{TP}{TP + FN} \tag{7}$$

$$specificity = \frac{TN}{FP + TN} \tag{8}$$

$$F_1 = \frac{2TP}{2TP + FP + FN} \tag{9}$$

$$precision = \frac{TP}{TP + FP} \tag{10}$$

ROC-AUC, defined as the area under the *sensitivity* – (1 – *specificity*) curve and PR-AUC, defined as the area under the precision-recall curve (recall is another word for sensitivity). Here, a positive case is defined as p ≤ 0.05 in MEP t-tests or ratio ≥ 1 in the LMFP ratios. While accuracy is used to assess the performance using all predictions from the model, sensitivity and specificity assess the performance of all positive and negative cases correctly identified as positives and negatives, respectively. Precision assesses the percentage of all cases predicted as positive being truly positives. $F_1$ score is equivalent to the harmonic mean of precision and sensitivity and is especially useful as a single metric for detecting uneven performance between precision and sensitivity.

10 repetitions of 5-fold stratified cross-validation were used to assess the performance for the model selection. The 95% confidence intervals were estimated using the student's t-distribution using the sample mean and standard deviation of 50 folds. All models with sensitivity or specificity under 0.60 are excluded for the model selection and model with the highest ROC-AUC was used for model selection during cross-validation. The model selected from cross-validation is then retrained on the entire training set and its performance tested on an external validation set. In this case, bootstrapping was used to compute the 95% confidence intervals, where the external validation set was re-sampled 2000 times, with each re-sampling set stratified to the class proportions of the original sample.

Details and examples on feature grouping and model generation (Fig 5) can be found in S1 Text.

## Supporting information

**S1 Text.**
(DOCX)

**S1 Fig. Model Selection (also called internal validation) Performance of LMFP Ratios Cross-Cohort Experiment for the selected feature set (coarse-graining multiscale entropy of the left motor region with LFMP ratios based on the 100–131 ms window).** 9 different classifiers are shown (left to right: logistic regression with Lasso (L2) regularization, linear discriminant analysis with inner cross-validation for hyperparameters, linear discriminant analysis with Ledoit-Wolf covariance matrix estimator, linear discriminant analysis with oracle approximating shrinkage, nearest shrunken centroid using Manhattan distance, nearest shrunken centroid using Euclidean distance, Gaussian naïve Bayes, Gaussian naïve Bayes with prior class proportion, and decision tree with maximum depth of 2 nodes). The vertical error bars represent 95% confidence intervals and the dark horizontal bars within each vertical bars represent theoretical chance levels. 7 different metrics are assessed: accuracy (blue), sensitivity (orange), specificity (green), F1-score (red), ROC-AUC (purple), PR-AUC (brown) and precision (pink).
(TIF)

**S2 Fig. Same as S1 Fig but for model validation (also called external validation).**
(TIF)

**S3 Fig. Internal validation (using cross-validation) results of the Cross-Subject Experiment for the LMFP Ratios.** The selected feature set is time-shifted multiscale distribution entropy for the central region using Log-Distance transformation and the selected labeling is based on the 85–115 ms window of the LMFP. 2 additional classifiers were tested for this experiment: support vector machine with linear or polynomial of 2nd order kernels. Metrics and legends are identical to S1 Fig.
(TIF)

**S4 Fig. External validation results of the Cross-Subject Experiment for the LMFP Ratios.** The feature set and labeling are identical to S3 Fig. Metrics and legends are identical to S1 Fig.
(TIF)

**S1 Table. Windows used for the left motor region of LMFP to categorize the modulation of cortical excitability as facilitation or suppression.** All times are shown in ms.
(XLSX)

**S2 Table. List of different measures of temporal complexity tested here. m represents embedding dimension and tau represents the embedding time delay.** For multiscale entropy, the embedding time delay range from 1 to 20.
(XLSX)

**S3 Table. List of ROIs and their EEG channels.**
(XLSX)

**S4 Table. Performance of model selection and model validation of Cross-Session Experiment.** Square brackets represent 95% confidence. Row a and b represent MEP Experiment. Row c and d represent LMFP Ratios Experiment. Row a and c represent model selection step whereas row b and d represent model validation step.
(XLSX)

**S5 Table. Coefficients of the final LDA OA model for the MEP Cross-Cohort Experiment.** Green box/text represents positive coefficients, red box/text represents negative coefficient. RQS stands for regression quality scores.
(XLSX)

**S6 Table. Gini Importance of the final decision tree in LMFP Ratios Cross-Cohort Experiment.** Higher indicates higher feature importance or contribution to the model.
(XLSX)

## Acknowledgments

The authors gratefully acknowledge the contributions of the participants, physicians and staff members.

## Author contributions

**Conceptualization:** Recep A. Ozdemir.

**Data curation:** Brice Passera, Recep A. Ozdemir.

**Formal analysis:** Matthew Herbert Ning, Brice Passera, Recep A. Ozdemir.

**Funding acquisition:** Mouhsin M. Shafi.

**Investigation:** Matthew Herbert Ning, Mouhsin M. Shafi, Recep A. Ozdemir.

**Methodology:** Matthew Herbert Ning, Brice Passera.

**Resources:** Mouhsin M. Shafi, Recep A. Ozdemir.

**Software:** Matthew Herbert Ning.

**Supervision:** Haoqi Sun, Brandon Westover, Alvaro Pascual-Leone, Emiliano Santarnecchi, Mouhsin M. Shafi, Recep A. Ozdemir.

**Validation:** Matthew Herbert Ning.

**Visualization:** Matthew Herbert Ning.

**Writing – original draft:** Matthew Herbert Ning.

**Writing – review & editing:** Matthew Herbert Ning, Haoqi Sun, Brice Passera, Duygu Bagci Das, Brandon Westover, Alvaro Pascual-Leone, Emiliano Santarnecchi, Mouhsin M. Shafi, Recep A. Ozdemir.

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
