## [Decision Letter · Decision Letter 0]

21 Nov 2025

Complexity of Resting Cortical Activity Predicts Neurophysiological Responses to Theta-Burst Stimulation but Fails to Generalize: A Rigorous Machine-Learning Approach.

PLOS Computational Biology

Dear Dr. Ozdemir,

Thank you for submitting your manuscript to PLOS Computational Biology. After careful consideration, we feel that it has merit but does not fully meet PLOS Computational Biology's publication criteria as it currently stands. Therefore, we invite you to submit a revised version of the manuscript that addresses the points raised during the review process.

We look forward to receiving your revised manuscript.

Kind regards,

Marcus Kaiser, Ph.D.

Academic Editor

PLOS Computational Biology

Lyle Graham

Section Editor

PLOS Computational Biology

**Additional Editor Comments:**

In addition to the reviewer's comments, please address the concerns of data availability:

"The Github repo https://github.com/NoPenguinsLand/TEP_ML is not available as of 18 Nov 2025. Please provide a publically accessible repository link, preferably with a permanent, indexed backup using a Zenodo release. The authors state that data is made available under https://bdsp.io/ , but no concrete link to a dataset is given. Searching for the paper title, "MEP", "iTBS" or other keywords didn't lead me to the data. Please provide a permanent link to the dataset."

**Journal Requirements:**

1) Please provide an Author Summary. This should appear in your manuscript between the Abstract (if applicable) and the Introduction, and should be 150-200 words long. The aim should be to make your findings accessible to a wide audience that includes both scientists and non-scientists. Sample summaries can be found on our website under Submission Guidelines:

**Reviewers' comments:**

Reviewer's Responses to Questions

**Comments to the Authors:**

Reviewer #1: In this work, authors use ML methods to predict neurophysiological responses to iTBS session. The result show promising prediction performance, however, the overall quality and impact can be significantly improved by addressing the following concerns:

1. authors highlighted that most machine-learning (ML) studies have focused on modeling behavioral or clinical effects of repetitive transcranial magnetic stimulation (rTMS), the few studies examining neurophysiological outcomes, so the proposed work fills this gap. However, it is important to discuss more about why it would be important to do the neurophysiological outcomes prediction and its responding challenges. Missing those discussion is misleading since focusing on behavioral or clinical effects of rTMS helps to address the challenges in application, which is essential for many fields.

2. Although the model is doing neurophysiological outcomes prediction, but based on the methods, it is still a classification model. The manuscript should reflect this clearly and mention this as one of its potential limitations. In most ML studies about response prediction, the model should intend to predict the signal response as a continuous variable.

3. The details of the Schematic Diagram (Fig 2) need to be improved. It's unclear what would be the input and what would be the output of the model (especially the output, and how the output is defined).

4. In the method section, authors should include the sample size (the number of trials) used in the model and have a detailed introduction.

5. For the cross-fold validation, did the author use leave-one-subject out methods? The current compelling result might be a result of data leak, since it might just simply learning the subject information (i.e., predict based on recognizing the subject).

6. Is the label (y) of the model being balanced?

7. Authors should include a baseline model as a control and compare its result with the included ML method. If an fair chance (such as 50% for binary or 33% for 3-class) can be reported, then address point 6 and authors can report the pair chance as control (but even after addressing point 6, a baseline model is still preferred because of the fuzzy nature of neurophysiological outcomes).

8. The provided GitHub link is expired and the dataset link is not directing to the dataset itself but the hosting platform.

Reviewer #2: The review is uploaded as an attachment. We like the conceptual and empirical groundwork but think certain statistical choices are necessary to make it more interpretable and robust.

Reviewer #3: Summary.

This work discusses iTBS, a special form of TMS using short bursts of magnetic stimulation. The authors address the large variability in inter-subject responses found in the literature, restricting its efficacy in clinical application. In particular, it remains unclear how neurophysiological responses are linked to iTBS.

By introducing a novel inter-cohort, inter-session evaluation pipeline, the authors aim to model iTBS efficacy variability using neurophysiological responses. They further test their pipeline using a large range of statistical tools on a novel dataset they have recorded, and discuss at length how their results can help robustify the analysis of iTBS efficacy.

Review.

The work is well motivated and the analyses presented feature a comprehensive list of tools applied. The need for external validation, i.e. an independent test set, is a sine non qua in traditional ML benchmarks; I am therefore happy to see this applied to clinical recordings, where data is much more sparse. I commend the authors' rigorous approach to address the important issue of subject response variability, even more so if that means that results become non-significant. By doing so, this work highlights the importance of rigorous testing procedures, in order to accurately estimate the true clinical value of iTBS.

Overall, the paper is well written, but organization and presentation of the manuscript requires some work. I found several passages hard to comprehend, e.g. due to the large number of acronyms and unfamiliar terminology the authors use (see below).

With some revision, I will be able to recommend this work for publication.

Major points.

(Please note that I refer to the internal validation dataset as "train set" in ML lingo, and external validation set as "test set".)

- The authors introduce two splits into train/test sets for their data: the cross-session split (experiment 1) and cross-cohort split (experiment 2). Why are separate analysis methods used for each of the two splits (statistical tests vs. training classifiers)? The authors motivate this with concept drift (page 20 line 21), but it did not become clear why the two dataset splits are paired to a specific analysis method.

- Main critique: Performance on the test set is generally reduced, for both the cross-session (p20l14) and cross-cohort (p22l21) analyses, and the authors suspect that is due to concept drift. In a strict definition, concept drift is a change in the decision boundary *occurring over time*. I understand that this may explain the performance drop in the cross-session split, but I am puzzled how concept drift could occur in the cross-cohort split. Even if we don't adhere to the strict definition above, but instead one where p(y|x) changes but p(x) does not (p13l5), it is not exactly clear whether it is p(y|x), or p(x) or p(y) which differ between train and test sets in the cross-cohort experiment. This argument requires further careful examination, as it appears to be one of main results underpinning the message of the paper. I'd like to see the reason of the performance drop identified; concept drift can be singled out through data augmentation or synthetic data, while the statistical tests of experiment 1 can be applied to the dataset split of experiment 2 to test for covariate and label shift.

- This closely ties in with a question regarding the cross-cohort split: Splitting the train/test datasets by subjects instead of session makes sense, but (why) do the subsets need to consist of the two data collection cohorts (2018-2021, 2016-2019)? This may be a reason for the decrease in test performance; instead of concept drift, it may be an overall bias present within each data collection cohort. I would like to see a cross-subject analysis, with subjects sampled randomly from both data collection cohorts.

- For experiment 2, the authors test a large range of classifiers (9) using cross-validation on the train set and testing on the test set. I commend this comprehensive analysis. I do wonder however how the different classifiers compare to each other? I could not find results for the ones which were not winners (LDA OA / decision tree), except for a mention in the supplement (see below).

- In experiment 2, the best models chosen are chosen via highest ROC-AUC during cross-validation on the training set. This may not be the best metric to find a model which is able to generalize, as it only relies on data from cohort 1 for model selection. Do the winning models remain the same if models are selected by highest test performance or smallest drop in performance between training and testing?

- Furthermore, have the authors considered training a support vector machine? SVMs are less likely to overfit and may extract more underlying features from the data than LDA (given an appropriate kernel size). I would like to see results for an SVM. That may also clarify if overfitting or concept drift (see above) are at fault for decreased performance.

- While the overall aim of the work is clear, the presentation of individual steps is at times confusing. In particular, nomenclature should be de-cluttered, and the presentation clarified:

-- Several sentences are filled with acronyms, and introduce multiple concepts at the same time. Example: p13l10-13.

-- In the definition of the paper, what is a "model"? In usual ML lingo, a model is the algorithm/classifier which is trained. But fig. 2 implies that it is the combination of feature group, categorization method and algorithm/classifier. As this definition only occurs in this caption, it was hard to understand the manuscript on my first readthrough. Please clarify across the manuscript.

-- On a similar note, on page 12, the authors define the training data for the classifier: features x and response variables y. It should be stated more clearly that x is the input and y is the label/target to the binary classifier; furthermore, "response variable" only appears this one time in the whole manuscript, and it is not clear that these are supposed to be the "responder/non-responder" classes. Please clarify in text. Including this in Figure 2 also would help clear up the confusion.

-- p13l10: what are intraclass correlation coefficients?

-- The authors number their two experiments, 1 (cross-session) and 2 (cross-cohort), but these numbers are not referenced again at the appropriate points in the analysis (also, "cross-session" only reappears in the discussion, not results section).

- At several points, no explanation for experimental protocols is given. While the experimental parameters are likely well motivated by clinical experience, an outsider will benefit from briefly reporting the reasoning. Examples:

-- p10l1: "In both studies, single pulses TMS were administered at 120% of the RMT of the motor hotspot." How did you choose the value 120%?

-- p10l15: "In both studies, the iTBS protocol was applied to the motor hotspot at 80% of the AMT" How did you choose the value 80%?

Minor points.

- In ML lingo, "internal validation dataset" is known as "train set" (in this case, with cross-validation), and "external validation set" as "test set". It would be good to define this clearly in the manuscript.

- It took some time for me to understand that all classifications in this work are *binary*, using the labels "responder/non-responder" for MEP and "facilitation/suppression" for LMFP. It would be good to clear this up early, e.g. page 10 line 17. You may want to include a sketch of the training pipeline in fig. 2 (see below).

- p21l15-15: "The performances are above chance level for all metrics. However, the confidence intervals for all metrics contain chance-levels." This statement is contradictory. The authors probably mean to say that the *mean* performance is above chance level. If so, please correct.

- p21l21-22: "To address possible concept drift, we change the way dataset is split in order to include both initial test and retest sessions during training" I found this sentence to be misleading, as concept drift is not *addressed* (i.e., solved), but rather *examined*. (there is also a typo, a "the" missing before "dataset")

- Presentation of results for the cross-cohort split should be streamlined:

-- Tab. 1: The added value of this table, compared to Fig. 4, did not become apparent to me. Also, formatting of the table needs to be improved: confidence intervals should appear in the same line as the mean value. Instead of rows "a", "b" etc, denote which dataset (train/test) was used. For rows b and d, show differences in performance w.r.t. rows a, c.

-- It seems that the subsection "MEP Cross-Cohort Experiment" also repeats the numbers of Tab 1 and Fig 4. The same goes for the subsection "LMFP Ratios Cross-Cohort Experiment". If they do not add anything, I suggest removing the paragraphs citing data.

-- Move Fig 4 to be close to Tab 1.

- p16l5: It took me some time to understand what is meant by "sensitivity-(1-specificity) curve". Please typeset as an equation.

- p16l6: How is "recall" defined?

- p10l2: "spTMS", while understandable in context, is never defined.

- Figures 2 and 4 are not placed near to where they are referenced.

- Figures and captions need to be improved. Fig. 1 is great, but most other figures do not make good use of space. They can be improved by removing whitespace, increasing fonts, and presenting a clearer flow of information.

-- Please include vectorized graphics in all cases and homogenize font sizes of labels w.r.t. captions and main text.

-- Fig. 2: The presentation could be improved. Why is the ROI selection part of "model generation"? (this relates to what the authors define as "model", see above) The figure would benefit from a 3-split presentation: Feature selection, label selection, classifier selection. Further, the workflow of the selection pipeline would benefit from an arrow from one step to the next (one table in top right connecting to the one in bottom left). Further, for the tables, different instances are stacked (top right, bottom center-right), but for the classifiers, they are separated vertically. Also, while all other choices are represented by an "x", the classifier selection is represented with 3 different arrows. Finally, you may want to picture how "features" and "labels" are fed into the classifier during training.

-- Fig. 3 is very hard to parse, as it presents too much information, is too small, and pixelated.

-- Fig. 4: The meaning of the terms "model selection" and "model validation" could be made clearer in this context. IIUC, the only difference is whether the internal validation dataset (cross-validation on train set) or external validation set (test set) are used. Please clarify.

-- Fig. 4: Figure design can be improved but removing whitespace, increasing fonts. The classifiers (LDA OA/tree) should be prominently displayed instead of as x-tick marks.

- Implications of the section "Results: MEP and LMFP Cross-Session Experiments" in the Supplement are not clear to me. What is the difference to the winning models (Table 1 in manuscript)? Please disambiguate.

**Have the authors made all data and (if applicable) computational code underlying the findings in their manuscript fully available?**

The PLOS Data policy requires authors to make all data and code underlying the findings described in their manuscript fully available without restriction, with rare exception (please refer to the Data Availability Statement in the manuscript PDF file). The data and code should be provided as part of the manuscript or its supporting information, or deposited to a public repository. For example, in addition to summary statistics, the data points behind means, medians and variance measures should be available. If there are restrictions on publicly sharing data or code —e.g. participant privacy or use of data from a third party—those must be specified.requires authors to make all data and code underlying the findings described in their manuscript fully available without restriction, with rare exception (please refer to the Data Availability Statement in the manuscript PDF file). The data and code should be provided as part of the manuscript or its supporting information, or deposited to a public repository. For example, in addition to summary statistics, the data points behind means, medians and variance measures should be available. If there are restrictions on publicly sharing data or code —e.g. participant privacy or use of data from a third party—those must be specified.requires authors to make all data and code underlying the findings described in their manuscript fully available without restriction, with rare exception (please refer to the Data Availability Statement in the manuscript PDF file). The data and code should be provided as part of the manuscript or its supporting information, or deposited to a public repository. For example, in addition to summary statistics, the data points behind means, medians and variance measures should be available. If there are restrictions on publicly sharing data or code —e.g. participant privacy or use of data from a third party—those must be specified.requires authors to make all data and code underlying the findings described in their manuscript fully available without restriction, with rare exception (please refer to the Data Availability Statement in the manuscript PDF file). The data and code should be provided as part of the manuscript or its supporting information, or deposited to a public repository. For example, in addition to summary statistics, the data points behind means, medians and variance measures should be available. If there are restrictions on publicly sharing data or code —e.g. participant privacy or use of data from a third party—those must be specified.

Reviewer #1: **No:** The provided GitHub link is expired and the dataset link is not directing to the dataset itself but the hosting platform.The provided GitHub link is expired and the dataset link is not directing to the dataset itself but the hosting platform.The provided GitHub link is expired and the dataset link is not directing to the dataset itself but the hosting platform.The provided GitHub link is expired and the dataset link is not directing to the dataset itself but the hosting platform.

Reviewer #2: **No:** We could not access the code via the provided link to the GitHub repository (https://github.com/NoPenguinsLand/TEP_ML).We could not access the code via the provided link to the GitHub repository (https://github.com/NoPenguinsLand/TEP_ML).We could not access the code via the provided link to the GitHub repository (https://github.com/NoPenguinsLand/TEP_ML).We could not access the code via the provided link to the GitHub repository (https://github.com/NoPenguinsLand/TEP_ML).

Reviewer #3: **No:** The Github repo https://github.com/NoPenguinsLand/TEP_ML is not available as of 18 Nov 2025. Please provide a publically accessible repository link, preferably with a permanent, indexed backup using a Zenodo release.The Github repo https://github.com/NoPenguinsLand/TEP_ML is not available as of 18 Nov 2025. Please provide a publically accessible repository link, preferably with a permanent, indexed backup using a Zenodo release.The Github repo https://github.com/NoPenguinsLand/TEP_ML is not available as of 18 Nov 2025. Please provide a publically accessible repository link, preferably with a permanent, indexed backup using a Zenodo release.The Github repo https://github.com/NoPenguinsLand/TEP_ML is not available as of 18 Nov 2025. Please provide a publically accessible repository link, preferably with a permanent, indexed backup using a Zenodo release.

The authors state that data is made available under https://bdsp.io/ , but no concrete link to a dataset is given. Searching for the paper title, "MEP", "iTBS" or other keywords didn't lead me to the data. Please provide a permanent link to the dataset.

PLOS authors have the option to publish the peer review history of their article (what does this mean?). If published, this will include your full peer review and any attached files.). If published, this will include your full peer review and any attached files.). If published, this will include your full peer review and any attached files.). If published, this will include your full peer review and any attached files.

...

Reviewer #1: No

Reviewer #2: **Yes:** David MenrathDavid MenrathDavid MenrathDavid Menrath

(Joshua P. Woller)

Reviewer #3: No

**Figure resubmission:**

**Reproducibility:**



---

## [Decision Letter · Decision Letter 1]

23 Mar 2026

Dear Dr Ozdemir,

We are pleased to inform you that your manuscript 'Complexity of Resting Cortical Activity Predicts Neurophysiological Responses to Theta-Burst Stimulation but Fails to Generalize: A Rigorous Machine-Learning Approach.' has been provisionally accepted for publication in PLOS Computational Biology.

Please consider the comment of reviewer 1 on making data available to increase the impact of this article.

Best regards,

Marcus Kaiser, Ph.D.

Academic Editor

PLOS Computational Biology

Lyle Graham

Section Editor

PLOS Computational Biology

Reviewer's Responses to Questions

**Comments to the Authors:**

Reviewer #1: Authors have addressed most of the previous comments. In order to increase the impact of the current work (or just validate the findings), authors should still consider make the data publicly available (maybe not raw, but at least include model inputs and outputs). It is quite common in the ML-related work and it is relatively important since the current manuscript is arguing neurophysiology changes have low generalizability (poor validation outcome).

Reviewer #2: We thank the authors for addressing our comments and for providing the additional analyses requested in the previous round. We have no further major concerns and consider the manuscript suitable for publication. We note, however, that the decision boundary of α < .05 represents an arbitrary decision parameter and should therefore be explicitly acknowledged as a meta-parameter for the model building and comparison. While it is objective and reproducible, it is not inherently better or worse than a 0.1 or 0.01 cut-off, even though it is more conventional. Reproducibility and objectivity are not inherently incompatible with arbitrariness. We also note that our original suggestion concerned retaining the binary classification while excluding samples near the decision margin from the training data, rather than introducing a three-class classification. Finally, while the link between spectral features and neurocognitive functions is well established and known to the reviewers, these measures still rely on temporal averaging, whereas complexity measures can more directly exploit the time-series nature of the data.

Reviewer #3: The authors have comprehensively addressed my concerns.

**Have the authors made all data and (if applicable) computational code underlying the findings in their manuscript fully available?**

The PLOS Data policy requires authors to make all data and code underlying the findings described in their manuscript fully available without restriction, with rare exception (please refer to the Data Availability Statement in the manuscript PDF file). The data and code should be provided as part of the manuscript or its supporting information, or deposited to a public repository. For example, in addition to summary statistics, the data points behind means, medians and variance measures should be available. If there are restrictions on publicly sharing data or code —e.g. participant privacy or use of data from a third party—those must be specified.requires authors to make all data and code underlying the findings described in their manuscript fully available without restriction, with rare exception (please refer to the Data Availability Statement in the manuscript PDF file). The data and code should be provided as part of the manuscript or its supporting information, or deposited to a public repository. For example, in addition to summary statistics, the data points behind means, medians and variance measures should be available. If there are restrictions on publicly sharing data or code —e.g. participant privacy or use of data from a third party—those must be specified.requires authors to make all data and code underlying the findings described in their manuscript fully available without restriction, with rare exception (please refer to the Data Availability Statement in the manuscript PDF file). The data and code should be provided as part of the manuscript or its supporting information, or deposited to a public repository. For example, in addition to summary statistics, the data points behind means, medians and variance measures should be available. If there are restrictions on publicly sharing data or code —e.g. participant privacy or use of data from a third party—those must be specified.requires authors to make all data and code underlying the findings described in their manuscript fully available without restriction, with rare exception (please refer to the Data Availability Statement in the manuscript PDF file). The data and code should be provided as part of the manuscript or its supporting information, or deposited to a public repository. For example, in addition to summary statistics, the data points behind means, medians and variance measures should be available. If there are restrictions on publicly sharing data or code —e.g. participant privacy or use of data from a third party—those must be specified.

Reviewer #1: **No:** The data that has been shared does not sufficiently support a manuscript's claims.The data that has been shared does not sufficiently support a manuscript's claims.The data that has been shared does not sufficiently support a manuscript's claims.The data that has been shared does not sufficiently support a manuscript's claims.

Reviewer #2: None

Reviewer #3: Yes

PLOS authors have the option to publish the peer review history of their article (what does this mean?). If published, this will include your full peer review and any attached files.). If published, this will include your full peer review and any attached files.). If published, this will include your full peer review and any attached files.). If published, this will include your full peer review and any attached files.

...

Reviewer #1: No

Reviewer #2: **Yes:** David MenrathDavid MenrathDavid MenrathDavid Menrath

Reviewer #3: **Yes:** Kevin MaxKevin MaxKevin MaxKevin Max

---

## [Editor Report · Acceptance letter]

PCOMPBIOL-D-25-02024R1

Complexity of Resting Cortical Activity Predicts Neurophysiological Responses to Theta-Burst Stimulation but Fails to Generalize: A Rigorous Machine-Learning Approach.

Dear Dr Ozdemir,

I am pleased to inform you that your manuscript has been formally accepted for publication in PLOS Computational Biology. Your manuscript is now with our production department and you will be notified of the publication date in due course.

With kind regards,

Anita Estes
